# Successful retrieval of competing spatial environments in humans involves hippocampal pattern separation mechanisms

Colin T Kyle[1], Jared D Stokes[1,2], Jennifer S Lieberman[1], Abdul S Hassan[1], Arne D Ekstrom[1,2]*

[1]Center for Neuroscience, University of California, Davis, Davis, United States; [2]Department of Psychology, University of California, Davis, Davis, United States

**Abstract** The rodent hippocampus represents different spatial environments distinctly via changes in the pattern of "place cell" firing. It remains unclear, though, how spatial remapping in rodents relates more generally to human memory. Here participants retrieved four virtual reality environments with repeating or novel landmarks and configurations during high-resolution functional magnetic resonance imaging (fMRI). Both neural decoding performance and neural pattern similarity measures revealed environment-specific hippocampal neural codes. Conversely, an interfering spatial environment did not elicit neural codes specific to that environment, with neural activity patterns instead resembling those of competing environments, an effect linked to lower retrieval performance. We find that orthogonalized neural patterns accompany successful disambiguation of spatial environments while erroneous reinstatement of competing patterns characterized interference errors. These results provide the first evidence for environment-specific neural codes in the human hippocampus, suggesting that pattern separation/completion mechanisms play an important role in how we successfully retrieve memories.

**\*For correspondence:** adekstrom@ucdavis.edu

**Competing interests:** The authors declare that no competing interests exist.

## Introduction

Place neurons (e.g. "place cells") in the rodent hippocampus preferentially fire in a particular spatial location (*O'Keefe and Dostrovsky, 1971*), the combination of which provide a neural code for that spatial environment (*O'Keefe and Nadel, 1978*; *Wilson and McNaughton, 1993*; *Muller and Kubie, 1987*). The collection of active place cells in an environment is thought to serve as a "cognitive map," providing a spatial framework for both navigation and memory more generally (*O'Keefe and Nadel, 1978*; *Redish, 1999*; *Buzsáki and Moser, 2013*). Two fundamental properties of place cells are their stability (*Thompson and Best, 1990*; *Hill, 1978*) and their environmental specificity, also known as "remapping" (*Muller and Kubie, 1987*). Without reliable recapitulation of the ensemble of place cells representing a specific "map," spatial memory is impaired (e.g., *Kentros, 1998*; *Morris et al., 1986*; *McHugh et al., 2007*). Remapping, a form of reorganization of hippocampal "maps" for different environments, is theorized to be a fundamental mechanism to navigation and memory more generally. However, the exact link between memory performance and remapping has yet to be fully established (*Jeffery et al., 2003*; *Colgin et al., 2008*). In humans, invasive recordings from the hippocampus have demonstrated place-coding neurons in single environments (*Ekstrom et al., 2003*; *Jacobs et al., 2013*; *Miller et al., 2013*). Additionally, the human hippocampal formation is important to episodic memory more broadly (*Spiers et al., 2001*), with place cells activating during item recall (*Miller et al., 2013*) and several studies demonstrating the

**eLife digest** How do we remember the different places that we have visited during our day? Studies of brain activity in rats suggest that each place a rat visits is represented by a different pattern of activity in a region of the brain called hippocampus. However, it is not clear what role this "spatial remapping" plays in the formation of memories in humans.

Kyle et al. used a technique called functional magnetic resonance imaging (fMRI) to investigate how our brain represents the different places we've visited, and how this is linked to how well we can remember these locations. For the experiments, human volunteers played a video game where they visited four virtual environments in a different order. Later, the volunteers were asked to remember details about the virtual environments they had visited while their brain activity was monitored using fMRI.

The experiments show that in order to remember distinct locations – even if they have some features in common – the hippocampus produces patterns of activity that have very little overlap with each other. This process is termed "pattern separation". Sometimes our memory confuses different locations, which could be due to a failure of the brain to distinguish between the patterns of brain activity that represent these locations.

Kyle et al.'s findings provide the first evidence for spatial remapping in the human hippocampus and its importance in forming memories of locations. The next steps are to find out how many different environments our brain might be able to store at the same time, and to identify factors that could aid in our memory for spatial locations.

ability to decode both location and episode-specific details from hippocampal fMRI blood-oxygen-level-dependent (BOLD) patterns (*Guterstam et al., 2015*; *Hassabis et al., 2009*; *Chadwick et al., 2010*). Whether the human hippocampus represents one spatial environment as either the same or different from another, however, – a cornerstone of the idea that the hippocampus may compute spatial "maps" as part of a larger role in processing memories—remains unknown and untested.

In addition to serving as a basic marker of memory, the environmental specificity of the hippocampus is thought to elucidate critical theoretical mechanisms of hippocampal function known as pattern separation and completion. These processes were predicted by early computational models and are thought to account for the memory interference errors commonly encountered in memory research and our everyday lives (*Yassa and Stark, 2011*; *Marr, 1971*; *Kohonen, 1977*; *Hunsaker and Kesner, 2013*). This theory states that pattern separation is a process that makes memories neurally distinct during memory storage and pattern completion a process by which memories are retrieved from a neural cue. Pattern separation and completion are thus thought to be important complements to each other (*Yassa and Stark, 2011*). Theoretical models and several empirical findings additionally suggest that CA3/DG and CA1 subfields mediate pattern separation and completion in the hippocampus (*Guzowski et al., 2004*; *Leutgeb et al., 2007*; *Leutgeb, 2004*; *Bakker et al., 2008*). Yet exactly how these findings relate to human spatial memory remains unclear.

Pattern completion is thought to rely on neural "attraction" between the cues that precede recall and stored representations, therefore allowing the cue to trigger re-instantiation of the full memory (*Rolls, 2010*). This property of attraction has the important implication that memories that are neurally similar will compete, producing interference in the case that the incorrect memory wins this competition (*Colgin et al., 2008*; *Shapiro and Olton, 1994*). Theoretical models, therefore, postulate the central importance of pattern separation as critical to making memories less similar and thus avoiding interference due to neural attraction. Alternative accounts of memory interference, however, instead argue against a pure pattern separation/completion based account in favor of a model which posits inhibition of interfering memories from executive control regions during memory recall (*Anderson, 2003*). This account instead predicts that similar representations can co-exist, but can be selected, maintained, and strengthened by executive control centers during memory retrieval. Therefore, a definitive neural link between behavioral interference, neural pattern separation, and spatial remapping is necessary to resolve this debate and clarify the function of the hippocampus in memory.

The aims of this study, thus, were three fold. The first was to examine whether humans also recapitulate neural codes for the same environment as well as bifurcate codes for different environments using fMRI and a multivariate pattern analyses. A second and critical test of whether remapping occurs in humans, however, is whether situations involving highly interfering spatial contexts can produce remapping failures (e.g., *Skaggs and McNaughton, 1998*; *Spiers et al., 2015*), and if so, what neural mechanisms characterize these errors. A final goal was to provide a link between behavioral measures of environment knowledge and neural measures of spatial remapping in humans.

## Results

To determine whether the human hippocampus also contains environment-specific neural codes, participants first explored four cities with varying levels of shared spatial context. Two cities (Cities 1 & 2) involved the same stores arranged in the same geometry, but with two store locations swapped (also termed the similar cities). A third city (City 3) involved the same stores as Cities 1 & 2, but arranged in a novel geometry and therefore at novel locations (interference city). Finally, City 4 involved a completely novel set of stores and geometrical arrangement (distinct city; see *Figure 1a*, Materials and methods). Following each round of navigation, participants drew maps of the environment to ensure that they accurately encoded spatial configuration details (*Figure 1b*, see Materials and methods). Following navigation and map drawing of all cities, participants entered the scanner where they performed two retrieval blocks per city (*Figure 1c*). Participants were instructed to recall a specific city during each retrieval block, with each trial involving judgments of the relative distances between stores (please see Methods for further details).

To better understand the extent to which the different cities involved competing representations, we compared learning rates of city-specific map-drawing performance in a separate behavioral study (*Figure 1—figure supplement 1–2* and Materials and methods). We predicted that City 3 would experience slower learning relative to Cities 1 & 2 because all of City 3's store locations were in conflict with store locations from Cities 1 & 2. Cities 1 & 2, on the other hand, had only 2 conflicting store locations, which could be learned via a simple swap. We found that the slowest learning did occur for the interference city (City 3) as well as the greatest confusion with the similar cities (Cities 1 & 2). In contrast, transitioning between the similar cities (e.g., City 1 to City 2) resulted in little detrimental effect on learning; in fact, learning one facilitated the learning of the other. Finally, no other cities facilitated learning of the distinct city (City 4) nor did learning the distinct city interfere with learning any other city. These findings suggested that City 1 and 2 (similar cities), despite being most similar, were easily distinguished from one another and that City 3 involved a representation prone to interference from similar cities (thus called the inference city). City 4, in contrast, did not interact with the other cities (distinct city) due to its novel stores and spatial geometry (*Newman et al., 2007*), allowing it to be readily distinguished from the other three cities.

We then tested whether participants' accuracy during retrieval of spatial distances of landmarks within the different cities (i.e., is store X closer to store Y or store Z?). Participants performed well above chance on spatial retrieval of all four cities (*Figure 1d*, single sample t-test against chance performance: $t(18) > 4.7$, $p < 0.0002$). A one-way repeated-measures ANOVA revealed differences in performance as a function of city ($F(3,54) = 20.9$, $p < 0.001$), which was driven by significantly lower retrieval performance on the interference city than all other cities ($t(18) > 4.65$, $p < 0.0002$). No other cities differed from each other in terms of performance, confirming the results of our navigation and map data analysis suggesting that retrieving information from the interference city (City 3) resulted in a tendency to confuse traces with those used in Cities 1 & 2. It is important to note that even though Cities 1 & 2 involved similar representations (with the difference being two swapped stores), performance on these two cities could not be explained by using the same responses between the two cities or the same responses and guessing on the swapped stores (see Materials and methods). These findings support the idea that participants were nonetheless using at least partially non-overlapping representations to retrieve details from Cities 1 & 2.

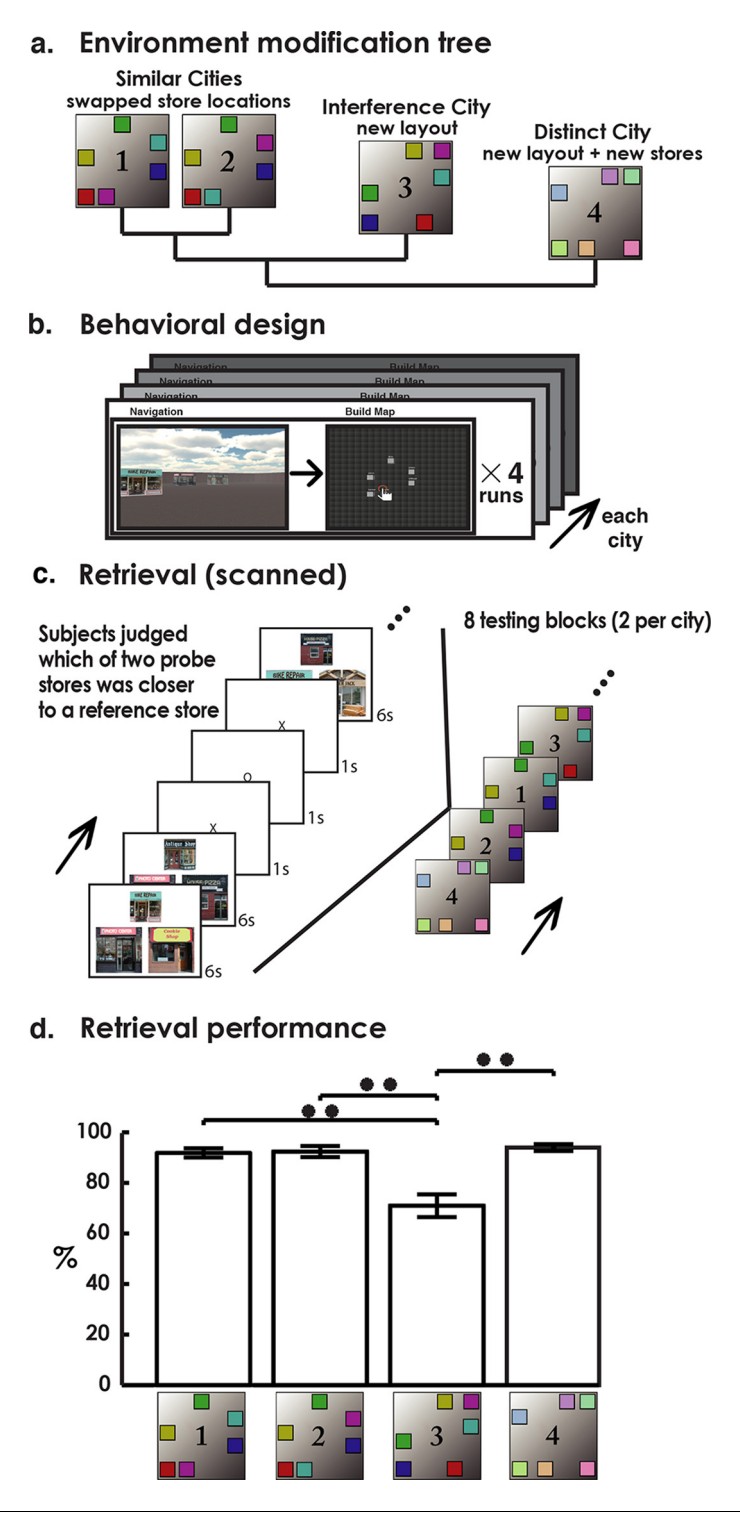

**Figure 1.** Experimental design and performance. (a) Depiction of contextual modifications between environments. Each colored box represents a different target store. Cities 1 & 2 (similar cities) are identical aside from swapped position of stores (purple and teal). City 3 (interference city) shares the same stores as similar cities but in a novel layout. City 4 (distinct city) has a novel layout and stores. (b) During encoding participants completed 4 rounds of navigation and map drawing of each city. (c) Retrieval consisted of 8 blocks of city-specific distance judgments. (d) Retrieval accuracy demonstrates lower performance on city 3. **p<0.01
*Figure 1 continued on next page*

*Figure 1 continued*

The following figure supplements are available for figure 1:

**Figure supplement 1.** Map drawing learning curves.

**Figure supplement 2.** City transition map scores.

## Classification of city-specific retrieval patterns in the hippocampus demonstrates successful decoding of all spatial contexts except the interfering environment

Our first and most basic prediction was that our human participants, analogous to remapping the rodent, would exhibit hippocampal voxel patterns that could uniquely identify each spatial environment. To address this prediction, we performed a searchlight classifier throughout the MTL (see Materials and methods for details and *Figure 2c*). This approach allowed us to naively identify MTL regions where voxel patterns carried city specific information. Our inclusion of the interference city into this analysis allowed us to address additional questions of pattern separation/completion. For instance, if the reduced retrieval performance of the interfering city could be attributed to insufficiently separated neural patterns where models predict neural competition at retrieval, the classifier should disproportionally misclassify the interference city as one of the similar cities (Cities 1&2) but not the distinct city (City4).

The pattern classifier correctly identified three of the four cities at levels above chance, revealing a cluster in hippocampal regions left CA3/DG and CA1 that significantly classified city identity (*Figure 3a–c*). Analyzing this cluster in a one-way repeated-measures ANOVA, with classifier performance of each city as a separate factor, revealed significant differences between cities (*Figure 3C*, $F_{(3,54)} = 12.9$, $p<0.001$). Testing each city's classifier performance against chance revealed that the classifier performed above chance on all cities except the interference city (Cities 1, 2, & 4 above chance: $t(18) > 3.2$, $p<0.006$, two-tailed). Conversely, interference city classification performance was consistently below chance levels ($t(18) = -3.2$, $p = 0.006$, two-tailed), despite overall classification (across all cities) being well above chance (*Figure 3C*, $t(18) = 5.6$, $p = 2 \times 10^{-5}$, two-tailed). We note that within our search space, which included the hippocampus, parahippocampal, fusiform, lingual, and inferior gyrus, only this cluster in CA3/DG and CA1 exceeded the family-wise error rate cluster size correction (see Materials and methods). Also, a control analysis that altered the classifier training protocol produced similar results suggesting these results were robust to differing analysis approaches (*Figure 3—figure supplement 1*, see Materials and methods for more details). This finding confirmed our prediction that, like the rodent, the human hippocampus contains environment specific representations in the CA1 and CA3/DG subfields for the environments that were most easily retrieved (Cities 1, 2, & 4). Additionally, the below chance classifier performance for City 3 and close resemblance of the retrieval accuracy and classifier accuracy seemed to indicate that low behavioral performance on City 3 may have related to lower levels of voxel pattern remapping for this environment.

To further explore the idea that low classification **levels** of the interference city (City 3) were due to the competing representations of the similar cities, we inspected interfering city classification results. Here, we predicted that the classifier would misclassify the interference city trials as either City 1 or City 2 on more than 50% of trials (chance level). This would be consistent with our behavioral results, which indicated that the interfering city was most often confused with Cities 1 and 2 during learning (see Materials and methods). This would potentially support a pattern separation/completion account of hippocampal remapping of spatial memory errors. As predicted, interference city trials were incorrectly labeled as one of the two similar cities at levels well above 50% ($t(18) = 2.7$, $p = 0.01$, two-tailed, *Figure 3c*). This suggested that a disproportionate amount of interference city trials resembled the similar cities. In contrast, trials for Cities 1 & 2, on which retrieval performance was well above chance, were incorrectly classified as City 3 significantly less than chance ($t(18) = -8.6$, $p<0.001$ corrected; see *Figure 3—figure supplement 2*). Overall, these findings are consistent with the idea that retrieval errors on City 3 could be attributed to, at least in part, insufficient differentiation of neural patterns from Cities 1&2.

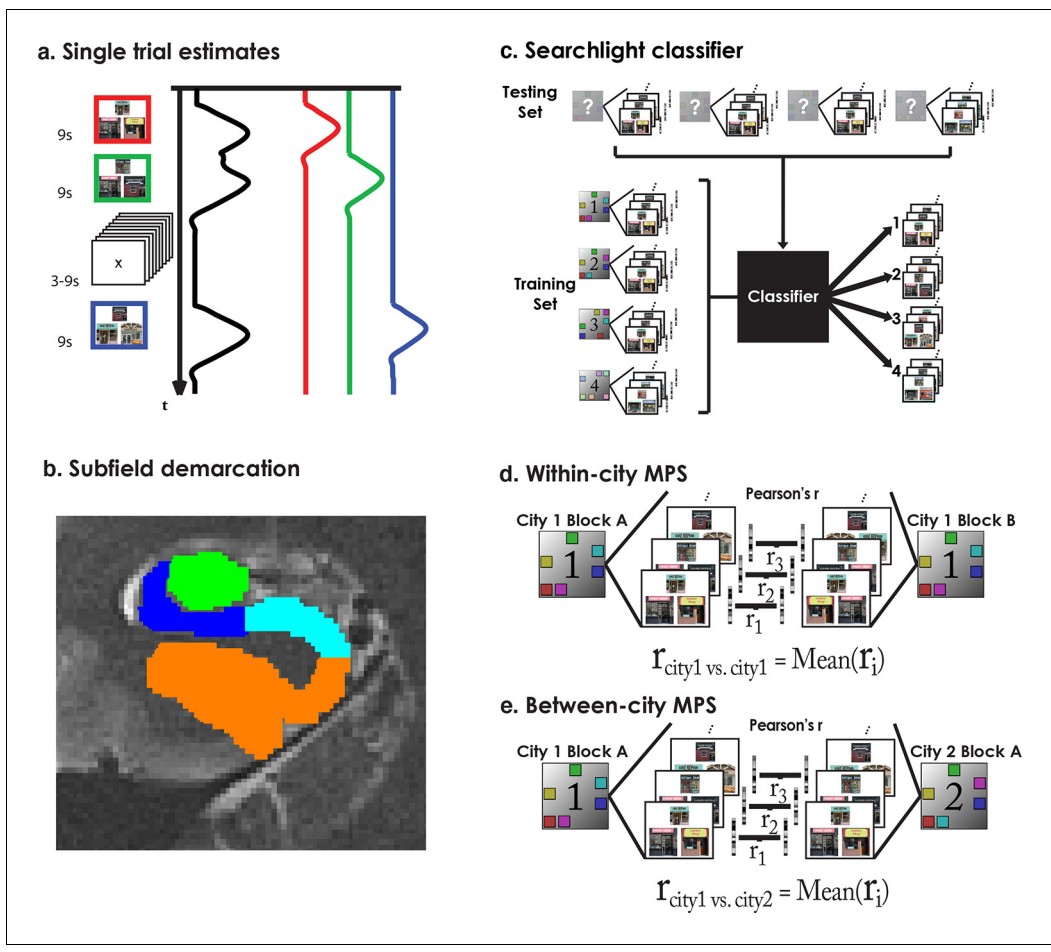

**Figure 2.** Analysis methods. (**a**) Single trial parameter estimates were generated by building a single model with a separate regressor for each trial. (**b**) Subfields were demarcated manually to create separate ROIs for CA3/DG, CA1, Subiculum, and PHG. (**c**) The searchlight classifier was trained using single trial estimates from half of the retrieval blocks and tested on the remaining retrieval data. Training/testing was repeated for all searchlight spheres in each subjects MTLs, creating subject specific statistical maps. (**d**) Within-city similarity was assessed for each ROI by extracting the trial parameter estimates from the subfields and correlating between matched trials of a city's "A" and "B" retrieval blocks. (**e**) Between-city similarity was calculated consistent with within-city similarity.

The following figure supplement is available for figure 2:

**Figure supplement 1.** Snapshot of virtual environment.

A second critical prediction from the pattern separation/completion account would suggest that better individual performance on the interference city should be attributable to more distinct and therefore more readily classified representations of this city. Therefore, we also examined the link between interference city classifier performance and participant retrieval performance by seeing if the two measures were correlated. We found that interference city retrieval and classifier performance were significantly correlated (r(17) = 0.61, p = 0.006, *Figure 3d*), a result which persisted even when matching the number of classifier training trials for each city (*Figure 3—figure supplement 1*). The link between participants who performed better on the interference city and those that had more readily classified neural representations of the interference city thus suggested that our best performing participants had neural representations that were more differentiated from each other than those of poorly performing participants. Together, the findings from the searchlight classification analysis support the idea that the human hippocampus exhibits environmental specificity. Further, the low classification performance of the interference city, and correlation between

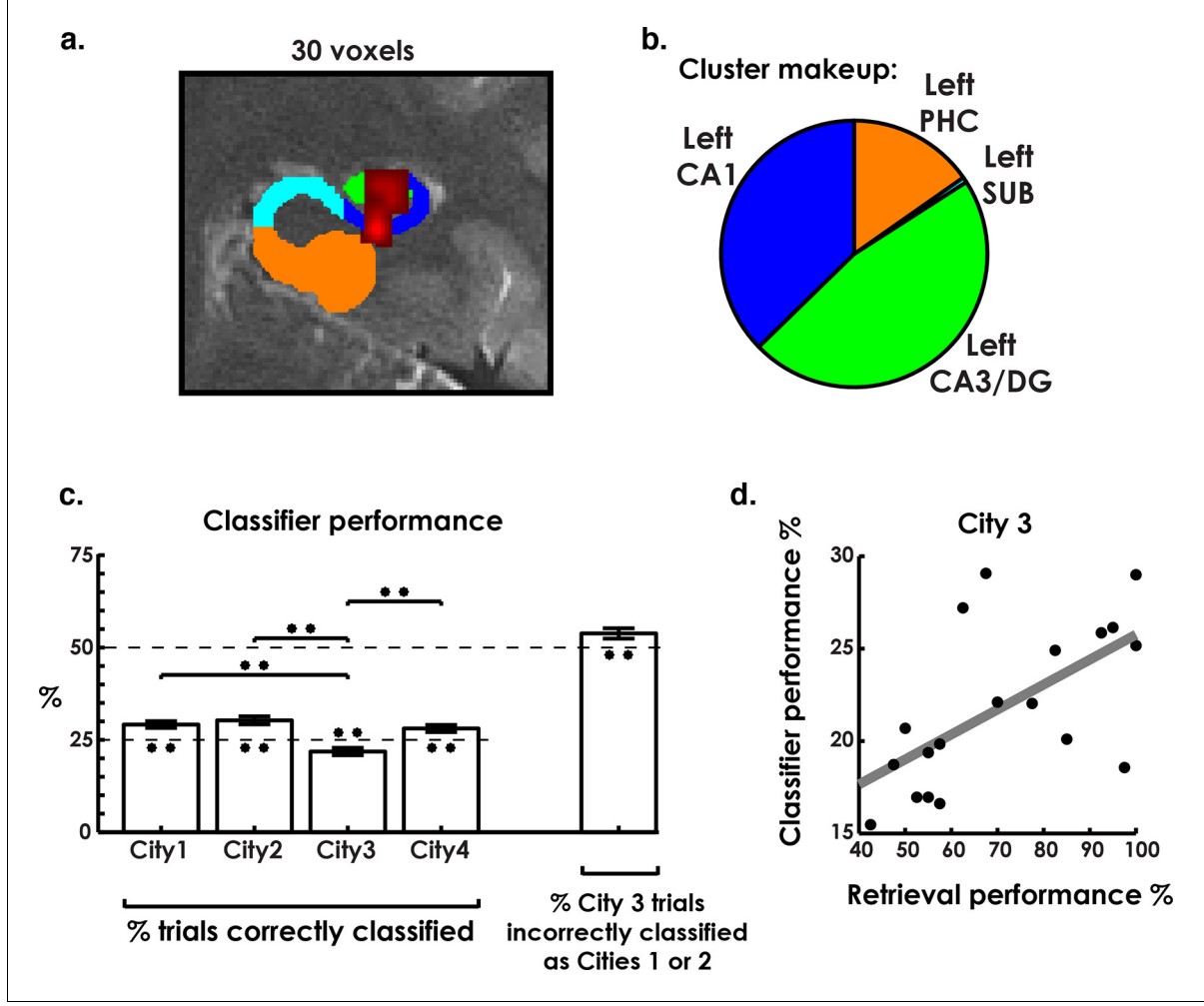

**Figure 3.** Environment classification. (**a**) City classification searchlight revealed a cluster of above chance classification performance throughout much of left CA3/DG and CA1. (**b**) Pie chart of distribution of voxels in the searchlight showing their predominance in CA3/DG and CA1. (**c**) Classifier performance of each city revealed above chance performance on cities 1, 2, and 4 and below chance performance on city 3. Further analysis of city 3 classification performance revealed above-chance misclassification of city 3 trials as cities 1 & 2. (**d**) City 3 (interference city) retrieval performance and city 3 classifier performance were positively correlated. *p<0.05, **p<0.01.

The following figure supplements are available for figure 3:

**Figure supplement 1.** Classifier trained with matched number of trials from each city.

**Figure supplement 2.** City 1 & 2 classification results broken down by correctly classified and incorrectly classified as each city.

behavioral and neural classifier performance, favor a pattern separation/completion based account of memory interference.

Although searchlight classifier analyses have advantages, in our case, the ability to naively identify regions within the human hippocampus exhibiting environment specific coding, they are not as well suited to hypotheses involving functional dissociations between subfields. For instance, searchlight clusters do not necessarily carry unique signals from different brain regions (*Woo et al., 2014*; *Etzel et al., 2013*). Pattern classification also cannot indicate whether neural codes are more similar within the same vs. between different environments, a cornerstone of spatial remapping findings in the rodent (*Wilson and McNaughton, 1993*; *Muller and Kubie, 1987*). We therefore employed a region-of-interest (ROI) based, multivariate pattern similarity (MPS) approach to 1) determine whether different human hippocampal subfields played different roles in spatial remapping and 2)

provide more specific alignment with findings from rodents indicating higher neural similarity for the same vs. a different spatial environment.

## ROI-based multivariate pattern similarity (MPS) voxel remapping suggests a functional dissociation of CA3/DG and CA1

As outlined in the prior section, our operational definition of remapping was voxel similarity within the same city (context reinstatement) and dissimilarity between different cities (remapping). To quantify this using multivariate pattern similarity (MPS), we created a voxel "remapping index" defined as within-environment similarity minus the average between-environment similarity (*Figure 2d–e*). Because our searchlight classifier analysis implicated CA1 and CA3/DG subfields in exhibiting voxel pattern remapping and remapping in the rodent is predominantly studied in CA1 and CA3/DG, here, we only include data from only CA1 and CA3/DG (for other subfield results, please see *Figure 4—figure supplement 1*). The results of this analysis are presented in *Figure 4a–c*. A 2x4 subfield by city repeated measures ANOVA revealed a main effect of subfield (F(1,18) = 6.93, p<0.02), and a marginal subfield by city interaction (F(3,16) = 3.14, p<0.06). This suggested that CA3/DG tended to exhibit more remapping but also that the pattern of results across cities differed for CA3/DG vs. CA1. Also of interest was whether each city remapping score was significantly

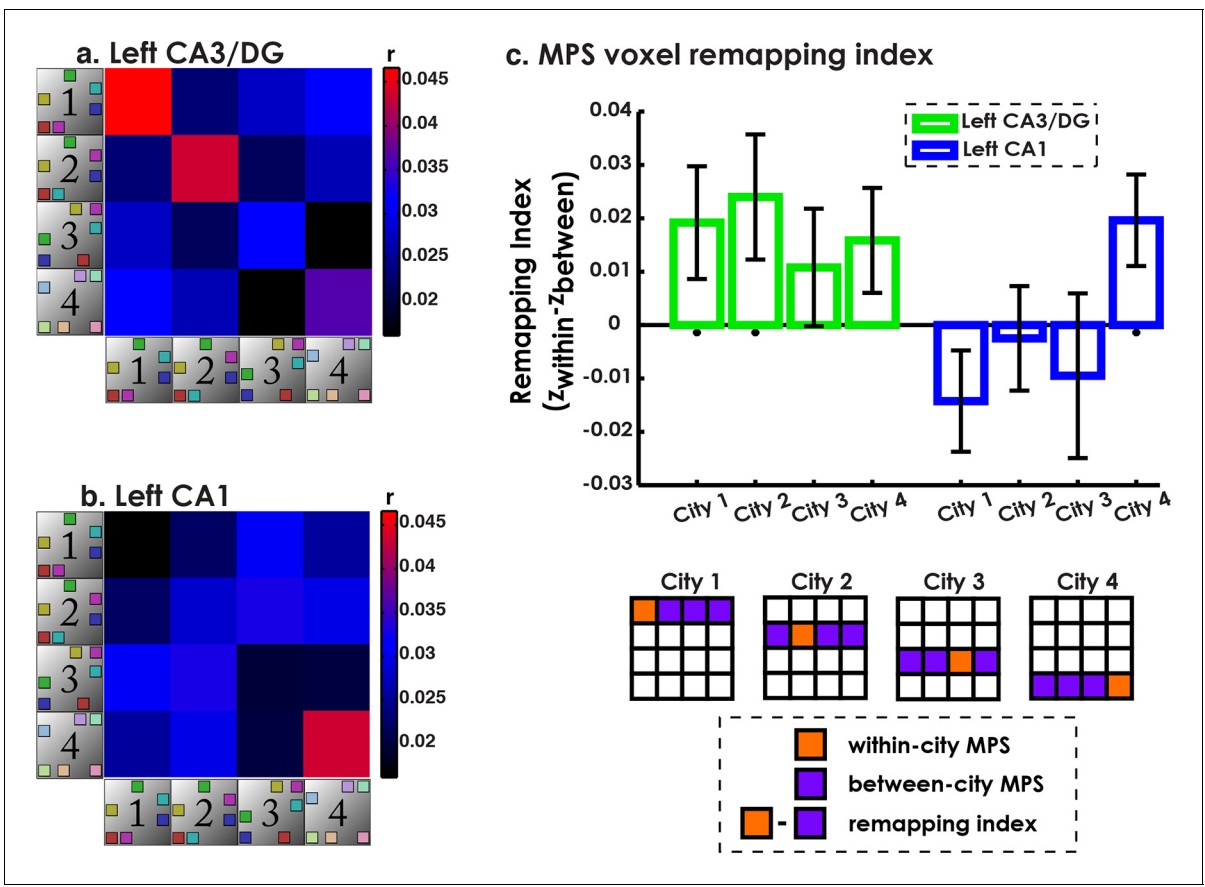

**Figure 4.** Multivariate pattern similarity analysis (MPS) ofenvironment similarity during retrieval. (**a**) Similarity matrix of all pairwise city MPS conditions in CA3/DG. Diagonal depicts within-city and off-diagonal depicts between city MPS conditions. (**b**) Same as (**a**) for CA1. (**c**) Voxel remapping index for CA3/DG (green) and CA1 (blue). Remapping index for each city was the z-transformed contrast between within city and average between cities MPS (see legend below). Left CA3/DG showed overall more remapping than CA1, with significant remapping for Cities 1 & 2 and marginally significant remapping for City 4. Left CA1 showed significant remapping only for City 4. *p<0.05.

The following figure supplement is available for figure 4:

**Figure supplement 1.** Cortical region MPS analysis.

different from chance. In left CA3/DG, remapping scores were significantly above zero for Cities 1 & 2 (all t-tests one tailed, t(18)>1.8, ps<0.05 corrected; see Materials and methods) and marginally above zero City 4 (t(18) = 1.6, p = 0.06 corrected). Thus, left CA3/DG MPS patterns indicated higher pattern similarity when participants retrieved spatial distances within Cities 1, 2, and 4 compared to the correlations of the patterns between different cities.

Left CA1, in contrast, had remapping scores that were significantly above zero for City 4 (t(18) = 2.29, p<0.05, corrected), but not for Cities 1-3 (ts<−0.25, p>0.6). Consistent with the searchlight classifier results, City 3 did not exhibit significant remapping in either CA3/DG or CA1 (ts<0.98, ps>0.17). This analysis further clarified the results of the searchlight classifier by suggesting that when characterizing entire subfields, more cities showed significant remapping effects in CA3/DG than CA1, but that CA1 did exhibit remapping for the most distinct city (City 4). Together, these findings suggested that CA1 and CA3/DG showed remapping for distinct city retrieval, while only CA3/DG showed remapping between similar cities. Overall, these findings also confirm the importance of pattern separation mechanisms to remapping in both human CA3/DG and CA1, an issue we consider with greater depth in the Discussion.

## Analysis of retrieval trials from the interference city suggests a partially unstable representation

Our searchlight classifier results suggested that correct trials tended to disproportionally resemble the similar cities but this effect was reduced for our best performing participants. This begged the question, if even correct interference city trials resembled the similar cities, did incorrect trials show even greater resemblance to the similar cities than the correct trials? If the representation of the interference city was unstable and easily attracted to the similar cities, then voxel patterns for incorrect interfering city trials should be highly correlated with voxel patterns of correct trials from the similar cities. Conversely, we would not expect to see high pattern similarity between similar cities and interference city trials if they were correctly answered. An important control comparison was included to make sure that this effect could be attributed to interference rather than a general property of incorrect retrieval. We would not expect the incorrect interference city trials to show similarity to the distinct city (because the distinct city was substantially different from Cities 1–3), and thus neither incorrect nor correct interfering city trials should have correlated voxel patterns with the distinct city.

To address these issues, we calculated MPS to compare correct and incorrect interference city trials with other city trials using matched visual stimuli (triads) (see Materials and methods), the results of which are presented in *Figure 5*. Voxel patterns in left CA3/DG (and right CA3/DG) on incorrect high interference city retrieval trials were significantly correlated with correctly retrieved voxel patterns in the similar cities (*Figure 5A*). Importantly, incorrect interference city trials were significantly more correlated with correct similar city (City 1 & 2) trials than were correct interference city trials with correct similar city or distinct city trials in CA3/DG (left bar greater than others, two-tailed t-test, t(18)>2.2, p<0.04). This effect was present in left CA3/DG (it was also present in right CA3/DG, two-tailed t-test, t(18)>3.7, p<0.001, *Figure 5B*, *Figure 5—figure supplement 1A*) but not present in CA1 nor any other subfield (t(18)<1.8, p>0.09, *Figure 4—figure supplement 1C*, *Figure 5—figure supplement 1B*). These results augment our searchlight classification results by demonstrating resemblance of incorrect interference city trials to the similar cities. Specifically, our findings support the idea that the unstable, weakly differentiated neural patterns of City 3 allowed the stable representations of Cities 1 & 2 to occasionally "outcompete" City 3 during retrieval. This in turn led to pattern completion of the wrong representation and selection of an incorrect response.

## Differences in univariate activations cannot account for city-specific representational contextual shifts

One potential issue with multivariate pattern analysis techniques such as classification and MPS is that they could be driven by simple effects related to increases or decreases in the BOLD signal at specific voxels and subfields rather than changes in distributed neural patterns (*Etzel et al., 2013*). It is also important to demonstrate that regions that carried multivariate information were recruited during the task by showing activation above baseline. To address these issues, we employed a simple univariate model comparing correct responses on each retrieval block against the baseline task

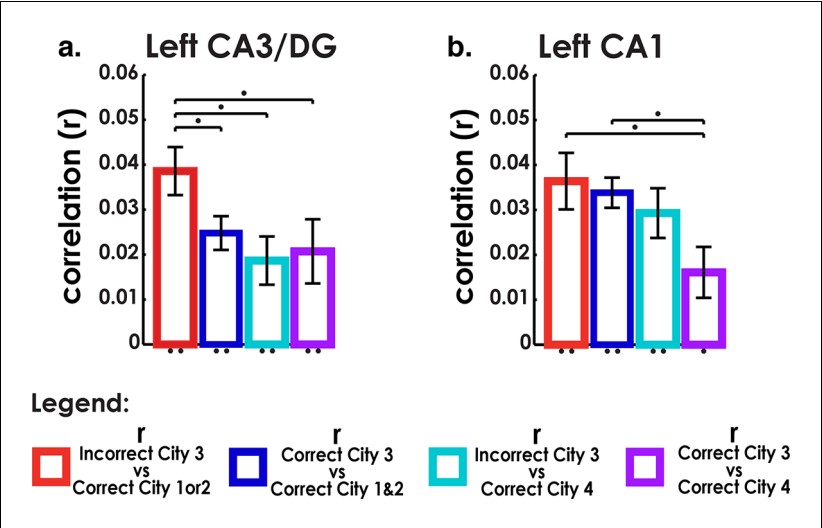

**Figure 5.** Analysis of incorrect and correct interference city trials. (**a**) Analysis of interference city trials reveals higher similarity between incorrect city 3 (interfering city) and correct city 1 or 2 trials than between correct city 3 and correct cities 1 and 2 trials in CA3/DG. Control comparisons suggest that this effect could be attributed to interference from cities 1 & 2. Left bar greater than all other bars t(18)>2.2, p<0.04. (**b**) CA1 did not exhibit similar behavior for incorrect vs correct between-city 3 comparisons. *p<0.05, **p<0.01.

The following figure supplements are available for figure 5:

**Figure supplement 1.** Right hemisphere hippocampal interference city MPS analysis.

**Figure supplement 2.** Empirical HRF plotted beside Canonical HRF convolved with 4 s boxcar function (average response time was 3.8 s).

(see Materials and methods). We found significant levels of activation across hippocampal subfields (average parameter estimates of CA1, CA3/DG, and subiculum, left hippocampus t(18) = 3.6, p = 0.002, right hippocampus: t(18) = 5.4, p = $3 \times 10^{-5}$; all t-tests two-tailed), confirming that the hippocampus was broadly activated by our task, consistent with our past work (*Kyle et al., 2015*; *Copara et al., 2014*; *Stokes et al., 2015*). We then tested whether MPS differences could be explained based on differences in univariate activation, which would challenge our findings of subfield specific changes in BOLD activation patterns (*Etzel et al., 2013*; *Davis et al., 2014*). To test this idea, we performed an 8x4 subfield (left and right CA1, CA3/DG, Subiculum, and PHC) by city repeated measures ANOVA on mean activation. This analysis revealed a main effect of subfield (F (7,126) = 29, p<0.001) driven by larger parameter estimates in PHC than hippocampus proper (t(18) >4.3, p<$4 \times 10^{-4}$). Neither of the remaining effects, however, (main effect of city and subfield by city interaction) were significant (F<1, p>0.5). We also specifically tested regions of interest CA3/DG and CA1 with a $2 \times 4$ subfield (left CA1 vs left CA3/DG) by city repeated measures ANOVA which revealed no significant effects (Fs<2, ps>0.15). These findings suggest that city-specific differences in univariate activation levels (i.e., greater activation to the distinct city than other cities) could not account for our overall pattern of results.

## Discussion

We believe that four novel components of our findings aid in understanding of human hippocampal function and its relation to memory processing. First, we extend environment specific coding to the human hippocampus using voxel-pattern based analyses. Using a searchlight classifier approach, which naively identified medial temporal lobe regions carrying city specific information, we found a cluster of voxels in CA3/DG and CA1 whose patterns decoded specific cities during retrieval. Next, using an MPS ROI approach which utilized all voxels from a subfield to characterize similarity within

and between cities using simple correlations, both CA3/DG and CA1 showed higher similarity within city than between cities. Although past studies in humans have confirmed the presence of location specific coding in the hippocampus (*Ekstrom et al., 2003*; *Jacobs et al., 2013*), measuring remapping typically requires a large number (>40) of simultaneously recorded cells (*Wilson and McNaughton, 1993*), which are difficult to obtain in most human studies. Additionally, although past fMRI studies in humans have suggested location-specific (*Guterstam et al., 2015*; *Hassabis et al., 2009*), distance-specific (*Morgan et al., 2011*), and episode-specific spatial coding within a single environment (*Brown et al., 2014*), demonstrating remapping between different spatial environments in particular has been elusive because altering the environment changes the visual scenes and trajectories experienced by the subject. Here, we dealt with this issue by minimizing visual confounds inherent in navigation by instead having subjects retrieve spatial distances from specific environments during retrieval. Thus, our findings suggest that indeed the human hippocampus contains neural codes that differentiate specific spatial environments.

Second, our results provide support for a pattern separation/completion based account of memory disambiguation. Here, we probed the neural underpinnings of both successful and unsuccessful disambiguation during memory retrieval. Retrieval of Cities 1, 2, & 4, which were more easily learned and retrieved, were shown to involve orthogonal voxel patterns, as demonstrated by a searchlight classifier and voxel similarity analyses. City 3, however, which contained repeated landmarks but in a novel arrangement from Cities 1 & 2, did not exhibit neural characteristics consistent with remapping or pattern separation, i.e. higher within than between city similarity or above chance classification. Rather, when attempting to classify City 3's correct retrieval trials, most trials were classified as City 1 or 2, although this effect was reduced for higher performing participants, suggesting that high performers exhibited more stable hippocampal patterns than low performers. Thus, one possible explanation for the poor performance on City 3 is that its neural patterns were insufficiently separated from those of Cities 1 & 2, resulting in a tendency to incorrectly pattern complete to stable representations of City 1 & 2. An alternative interpretation could be to attribute such errors to inhibition failure, for example, insufficient inhibition of City 1 & 2 representations by prefrontal cortex could lead to those being erroneously retrieved when attempting to retrieve City 3 (*Anderson, 2004*). The inhibition model, though, would not appear to predict low classification of correct City 3 trials and misclassification of these trials as Cities 1 & 2 trials since correct City 3 trials should involve trials in which traces from Cities 1 & 2 were successfully suppressed (*Aron et al., 2014*) and thus show no correlation with Cities 1 & 2 (see *Figure 5*). Furthermore, it is not clear how inhibition from higher cortical areas alone could lead to different patterns of suppression across the hippocampal subfields as prefrontal cortex projects primarily to subiculum and entorhinal cortex and not differentially to the CA fields, at least in non-human primates (*Goldman-Rakic et al., 1984*). Thus, our findings overall support the importance of pattern completion and separation, particularly in CA3/DG, to spatial remapping and appear less easily reconciled with an inhibition-based account.

A third important insight provided by our findings is a potential link between remapping-like mechanisms in humans, spatial learning, and rodent hippocampal remapping. The relationship between hippocampal remapping and behavior, however, remains unclear from the few studies to address both (*Colgin et al., 2008*). Part of the issue, as acknowledged in past such studies, is that it is difficult to assay whether a rat "knows" it is in a different environment or not, although dwell time and reversing direction may be important behavioral assays (*Spiers et al., 2015*). *Jeffery et al., (2003)* show that in a hippocampally dependent place-reward discrimination task, rodents perform only slightly worse after small environment modifications that induce global (~85% of cells) remapping (*Jeffery et al., 2003*), suggesting that remapping can occur quickly and have little negative effect on performance. *McHugh et al., 2007*, in contrast, demonstrated that dentate gyrus NMDA knockout mice experienced less hippocampal remapping between contexts and less behavioral discrimination (*McHugh et al., 2007*), suggesting that remapping is important to behavior. In the current study, we assessed map drawing performance after each round of spatial exploration, which provided a more direct link to the formation of a cognitive map during navigation. We found that maps of Cities 1 & 2 were easily learned because information could be readily transferred between the two cities. Later, when assessing neural patterns during retrieval, Cities 1 & 2 were shown to have mutually orthogonal hippocampal voxel patterns. In contrast, City 3's maps were less accurate and took more trials to acquire due to interference from Cities 1 & 2. During retrieval, performance was lower and hippocampal patterns were not orthogonal to those of Cities 1 & 2. However, we

found that participants who performed better on City 3 did show voxel patterns that were more readily differentiated from the other cities. Thus, our findings from Cities 1 & 2 appear consistent with the results of *Jeffrey et al. 2003* as we show remapping between Cities 1 & 2 despite the map acquisition data arguing for shared information between the two distinct representations. Our results for City 3, though, appear consistent with *McHugh et al., 2007*, with less remapping negatively impacting memory performance. Thus, we think our data provide a potentially important link between behavioral memory performance in humans and measures of remapping and pattern separation/completion.

Finally, our findings also provide important extensions and challenges regarding the function of the human hippocampal subfields in spatial context disambiguation. Specifically, when correctly retrieving spatial distances from two overlapping cities differing only in terms of two swapped stores, neural patterns were uncorrelated to each other and all other cities, an effect primarily shown in left CA3/DG. These findings support a role for CA3/DG in differentiation of competing spatial inputs, suggesting that this subfield in particular may be important for fine-grained discriminations amongst overlapping contexts as a part of a larger role in pattern separation/completion (*Yassa and Stark, 2011*). We also found that CA3/DG pattern remapping was (marginally) significant for City 4, suggesting that it differentiated the distinct city as well. These findings are consistent with the idea of CA3/DG as a universal pattern separator/completer, with failures linked to low performance on City 3 (*Yassa and Stark, 2011*). In contrast, pattern similarity in CA1 was higher when participants correctly judged spatial distances from a distinct city featuring novel landmarks and geometry compared to retrieval-induced patterns from the other cities. These findings appear somewhat inconsistent with models that suggest CA1 serves as a complement to CA3/DG in pattern separation/completion (*Guzowski et al., 2004*). Instead, our data appear to be more consistent with the emerging idea that CA1 may play a specific role in detection or representation of novelty (*Duncan et al., 2012*; *Larkin et al., 2014*), possibly acting as an important hub for integrating cortical input (*Stokes et al., 2015*; *Remondes and Schuman, 2004*). Although our study cannot provide specific insight beyond this speculation regarding the functional role of CA1, our findings suggest that its role in processing spatial contexts goes beyond a pattern separation/completion function defined by its position between CA3/DG and entorhinal cortex.

One potentially perplexing aspect of our findings is that we found similar degrees of remapping, and therefore putative neural pattern separation processes, for both the similar cities (Cities 1 & 2) and City 4, despite the fact that information content was significantly different for City 4 than Cities 1&2. Past studies, for example, have found that neural pattern differences may scale as a function of environment dissimilarity, with geometrically more distinct environments showing lower neural pattern similarity than more similar-shaped environments (*Stokes et al., 2015*; *Leutgeb et al., 2005*). In contrast to these two studies, though, we did not employ a continuous measure of environmental similarity and retrieval success hinged on being able to successfully maintain separate representations for the different environments but not necessarily tracking differences in the details of the different environments themselves. Thus, it is possible that our paradigm involved a more discrete form of pattern separation (*Yassa and Stark, 2011*) than would be needed in experiments involving a continuous change between environments. Consistent with the discrete pattern separation interpretation, studies of human episodic memory suggest that even overlapping episodes can be successfully decoded from multivariate patterns in the hippocampus, suggesting that retrieval of even very similar episodes can involve pattern separation-like processes (*Chadwick et al., 2010*). Thus, one possible interpretation of our results is that pattern separation in CA3/DG may not always scale with behavioral details and thus may be a more discrete process that depends on the exact demands required by encoding and retrieval.

In conclusion, while past studies in rodents suggest that hippocampal remapping might be one possible mechanism whereby memories are differentiated, no studies have demonstrated a comparable phenomenon in humans. Our findings thus provide several important insights about the human hippocampus: 1) the hippocampus contains unique codes about specific spatial environments, with these codes showing significant neural pattern similarity within the same spatial environment and low similarity between different spatial environments 2) when interference between spatial environments is high, pattern completion/separation processes may fail, resulting in difficulties discriminating competing cities 3) remapping, in humans at least, appears tightly linked with behavioral performance at discriminating different environments during retrieval 4) human CA3/DG appears to

play a fairly ubiquitous role in pattern separation/completion during retrieval of specific spatial environments while CA1 plays a more specific role in representing features of novel environments. Together, our findings provide new insight into how the human hippocampal circuit processes competing spatial information.

## Materials and methods

### Participants

Nineteen healthy individuals participated in the experiment (9 female) from the community surrounding University of California at Davis. All participants had normal or corrected-to-normal vision and were screened for neurological or psychiatric illness. This study was approved by the Institutional Review Board at the University of California at Davis. Written informed consent was obtained from each participant before the experiment.

### Task procedures

The study consisted of a learning session (not scanned) and retrieval session (scanned). During the spatial learning session, subjects played a video game where they learned four virtual environments on the computer in a randomized order. Each virtual environment consisted of six "stores" (for a snapshot, see *Figure 2—figure supplement 1*). Participants learned store locations by traversing the environment and then drawing a map of the store locations after visiting each store. This process repeated four times in the imaging study and six times in the behavioral study (see Materials and methods) before participants moved on to the next city. Traversals involved traveling to each store in the environment in a randomized order.

The layout of each environment is shown in *Figure 1a*. Similar cities (Cities 1 and 2) were the same except the locations of two stores were swapped in these environments. City 3 (called the interference city based on the behavioral results) contained the same stores as Cities 1 and 2, but in a novel layout and with novel ground and wall textures. Finally, City 4 (distinct city) contained a novel store set, a novel layout, and novel ground and wall textures. Novel ground and wall textures were used in Cities 3 and 4 to reduce interference in these cities based on the findings of *Newman, et al., 2007*. We instructed participants that there would be four cities, some involving repeated stores, and that they would need to distinguish each city from one another to successfully perform the retrieval portion of the task.

The retrieval portion took place in the scanner, where participants completed eight retrieval blocks (two per city). Each retrieval block consisted of ~4½ min of memory judgments pertaining to a single city. Before the start of each retrieval block, text and verbal confirmation informed participants of which city they would be retrieving next, followed by a 40 s refresher video. After completing a retrieval block, participants were permitted a brief break before moving on to the next block and thus a new city. The order of retrieval blocks was pseudo randomized with rules dictating that no city could be tested twice in a row and that each city must be tested once before a city could be repeated. Each block consisted of 20 trials of judgments of the relative distances of stores in that city. Each trial consisted of an image of 3 stores, one store on top and two below. Participants judged which of the two bottom stores was closer to the top store, and indicated their choice by pressing the corresponding key on an MR-compatible button box. A "one" response indicated that the lower left store was closer to the top store and a "two" indicated the lower right. Because Cities 1–3 shared the same stores, they also shared the same stimulus set, while city 4 contained a novel stimulus set. To control for effects of motor response, the position of the two bottom probe stores on each stimulus was swapped during "A" vs "B" retrieval blocks of the same city, effectively switching the correct button responses for the different sessions of the same city (see *Figure 2c*).

### Behavioral assay of spatial contextual shifts

To determine behaviorally whether participants employed similar, competing, or novel representations for spatial context, we tested each participant on how they encoded four different virtual cities (*Figure 1a*). This involved navigating one of the four cities and then drawing a map immediately afterward. The first two cities (Cities 1 & 2 involved the same stores and geometry with two locations swapped [similar cities]); the third city involved the same stores as cities 1 &2 but a novel geometry

and locations (interference city). City four involved a completely novel set of stores and geometrical arrangement (distinct city; see *Figure 1*, Materials and methods, and *Zhang et al. 2014*). We tested a total of 31 participants. Map drawing accuracy improved for all four cities as a function of navigation (*Figure 1—figure supplement 1*: City 1 & 2: Beta = 0.065, $F(1,170) = 34.95$, $p = 1.8 \times 10^{-8}$; second city 1 & 2: Beta = 0.025, $F(1,175) = 7.36$, $p = 0.007$; City 3: Beta = 0.058, $F(1,179) = 38.41$, $p = 4 \times 10^{-9}$; City 4: Beta = 0.033, $F(1,179) = 14.52$, $p = 0.0002$), suggesting that participants were able to form stable representations of each one. We next wished to address the extent to which acquiring a representation for one environment might enhance or impede acquiring a representation for another environment. This, in turn, would provide insight into the extent to which forming a representation of the different environments involved overlapping, different, or competing representations, as hypothesized. To address this issue, we computed the difference in map drawing scores during city transitions, in other words, the extent to which drawing a map on the last trial of learning interfered with drawing the map of a new city on the first trial (i.e., first drawn map of the new city – last drawn map of the previous city.

Comparing changes in map scores when transitioning from one city to another, we found a main effect of city transition (*Figure 1—figure supplement 2*, 1- Way ANOVA, $F(3,74) = 3.83$, $p<0.02$). Post hoc two-tailed t-tests demonstrated that transitioning to City 3 from City 1 or 2 resulted in significantly greater learning costs than transitioning from City 1 to City 2 or from City 2 to City 1 ($t(41) = 2.4$, $p<0.02$; $t(38) = 2,7$, $p<0.01$). Similarly, transitioning from city 3 to cities 1 or 2 resulted in significantly greater learning costs than transitioning to City 4 ($t(33) = -2.2$, $p <0.04$). This supports the assertion that City 3 representations interfered with cities 1 and 2. The only city to city transition that did not result in significant learning costs was the City 1 to City 2 transition (two-tailed t-test: $t(27) = -1.9$, $p = 0.07$; all other city to city transitions involved significant costs in map drawing performance (one tailed t-tests against zero, $ts>-3.3$, $ps<0.007$). This supports the idea that Cities 1 and 2 facilitated learning of each other. Transitioning to City 4 from any other city compared to transitioning from Cities 1 and 2, however, did not differ ($t(49) = 1.0$, $p = 0.3$). This suggests that City 4 was likely represented by a new representation entirely. Together, our findings support the idea that Cities 1 and 2 likely involved similar, largely overlapping representations (similar cities), City 3 involved a representation interfering with City 3 (interference city), and City 4 likely involved a novel non-overlapping, novel representation (distinct city).

## Behavioral analysis of city 1 & 2 swap trials

We designed our stimuli such that trials involving stores that swapped locations between Cities 1 and 2 were over-represented. For instance, 12 of 20 trials involved at least one swapped store between Cities 1 & 2 and 9 of these 12 had a different correct response in City 1 vs City 2. Therefore, a subject could score a maximum of 55% accuracy in City 2 based on knowledge of City 1 alone, and vice versa and all subjects were well above this threshold. Additionally, even if we assume that subjects correctly answer all trials involving the same response in Cities 1 & 2 and *guess* on trials involving different correct responses, we would expect a chance level accuracy of 77.5% for cities 1 & 2. All but 2 subjects had accuracy above 77.5% for *both* Cities 1 and 2 (one scored 77.5% on City 1 and 92.5 percent on City 2; the other scored 72.5 and 57.5 on Cities 1 & 2, respectively). Furthermore, taking the lower performance for Cities 1 and 2 for each subject and testing the result against 77.5%, subjects still performed significantly above chance ($t(18) = 5.4$, $p<0.0001$). Thus, performance on the swapped cities (Cities 1 & 2) could not be explained by a strategy involving using the same response on both cities.

## fMRI data acquisition, preprocessing, and parameter estimation for univariate analyses

We employed the same imaging sequences and preprocessing steps described in *Kyle et al., 2015* and *Stokes et al., 2015*. Imaging took place in a Siemens 64-Channel 3T "Skyra" scanner. High-resolution structural images were acquired employing T2-weighted turbo-spin echo (TSE) anatomical sequences (TR = 4200.0 ms, TE = 93.0 ms, FOV = 1.9 mm, flip angle = 139°, bandwidth = 199 Hz/pixel), involving a voxel resolution of $0.4 \times 0.4 \times 2$ mm. High-resolution functional echo-planar imaging (EPI: TR = 3000 ms, TE = 29 ms, slices = 36, field of view (FOV) = 192 mm, flip angle = 90°, bandwidth = 1462 Hz/pixel) involved a resolution of $1.6 \times 1.6 \times 2$ mm. Sequences were acquired

perpendicular to the long axis of the hippocampus. An additional matched-bandwidth sequence was acquired to aid in registration of the EPI sequence to the high-resolution scan (TR = 3000 ms, TE = 38 ms, slices = 36, FOV = 245 mm, flip angle = 90°, bandwidth = 1446 Hz/pixel). Each EPI sequence underwent band pass filtering, slice-timing, and motion correction in SPM8 before parameter estimation. Parameter estimation for univariate analyses used a canonical hemodynamic response function (HRF), and modeled all correct responses above baseline for each EPI sequence (*Friston et al., 1995*).

## Parameter estimation for multivariate analyses

Analysis of multivariate pattern similarity requires maximally orthogonalized hemodynamic response functions (HRFs) as collinearity can inflate MPS-related correlations (*Mumford et al., 2012*). Consistent with past work, we modeled each trial as a separate regressor (*Copara et al., 2014*; *Mumford et al., 2012*; *Rissman et al., 2004*) using finite impulse response (FIR) functions to model the average HRF to retrieval stimuli. This produced 10 parameter estimates for the first through the tenth TR after stimulus onset, corresponding to a 30 s long time course estimate of the HRF for each subject, block, and voxel (*Mumford et al., 2012*; *Mourão-Miranda et al., 2006*). This ensured the greatest ability to detect when spatial contextual retrieval might occur for the different cities but without selecting specific HRFs for different subjects or conditions. To select the HRF that explained the most variance for all subjects, sessions, and voxels, we employed independent component analysis decomposition using logistic infomax ICA (*Bell and Sejnowski, 1995*) and identified a single HRF component that explained 38% variance (shown in *Figure 5—figure supplement 2*). This HRF was then resampled using a cubic spline interpolation to match the 16 time-bin per scan default that SPM8 uses to build regressors.

## Subfield demarcation

Separate left- and right- hemisphere anatomical ROIs were manually traced (using FSLview) based on each participant's high resolution T2 as described previously (*Copara et al., 2014*; *Ekstrom et al., 2009*). Demarcated subregions included hippocampal subregions CA1, CA3/DG, Subiculum, and the extrahippocampal region parahippocampal cortex. We combined the CA3/DG subfield as finer distinctions cannot be made at the acquired resolution. MPS analyses were based on all voxels identified within ROIs.

## Classification analysis

We performed classification using the Princeton mvpa toolbox (*Detre, 2006*), with alterations to the code to allow three hidden layers and a searchlight across MTL subfields. The searchlight was performed as in our previous manuscripts (*Copara et al., 2014*; *Stokes et al., 2015*; *Kyle 2015*). Briefly, for each 31 voxel ellipsoid throughout each subject's MTL, we trained the classifier on one half of retrieval blocks (one block per city) and used the second half to test classification accuracy then swapped training and testing data. Two classifier training protocols were used. The first trained the classifier using all correct trials from one half of retrieval blocks. This method maximized the amount of training data but did not balance the number of trials used to train each city (*Figure 2c* and *Figure 3*). A second classifier training protocol used a random subset of correct trials from each city so that the classifier would be trained using the same number of trials from each city (*Figure 3—figure supplement 1*). Next, the average classifier performance for each searchlight position created a subject specific statistical map. Maps were warped to common space of a template subject using Advanced Normalization Tools (*AVANTS et al., 2008*). Finally, group-space maps were contrasted and clustered by t-value corresponding to alpha = 0.05. Permutation tests corrected for false positives by providing a corrected p<0.05 cluster size from the distribution of max cluster size of 1000 label-shuffled permutations. We note that a control analysis expanded search volume outside of the MTL to include fusiform gyrus and inferior temporal cortex, no clusters from these regions passed threshold.

## fMRI multivariate pattern analysis (MPS)

Pattern similarity analysis involves measuring the similarity of voxel patterns by calculating the correlation between parameter estimates of different trials within a common collection of voxels

(*Mumford et al., 2012*; *Kriegeskorte et al., 2006*). To measure pattern similarity, we identified trials that were correctly retrieved during two separate retrieval blocks. For within city MPS correlations were made between blocks of the same city and for between city MPS between blocks of different cities. MPS values measured the average r value between matching, correctly answered trial pairs for each participant and each subfield (*Figure 2c*). The bottom stores of each stimuli were swapped between different testing sessions of the same city, eliminating contributions to within-city MPS from the same motor response (*Figure 2d*). Because Cities 1–3 shared a common stimulus set, within and between city MPS could be calculated identically for these cities. The ability to match stimuli identity for within and between city MPS for Cities 1–3 allowed excellent control for visual aspects of the task as any differences in patterns could be attributed solely to retrieval environment. Because City 4 contained novel stimuli, necessitated by our behavioral testing to ensure that this city involved a new representation, between-city comparisons involving City 4 had no logical matching stimuli. Thus, instead of matching trials based on stimuli identity, all correctly retrieved stimuli from City 4 were correlated with all correctly retrieved stimuli from the other city for all possible pairwise combinations. A control analysis calculated between-city MPS forCities 1-3 using the same method as City 4, with all pairwise combinations of non-visually matched triads. This control analysis did not reveal any significant deviations to our effects and thus visual matching was maintained when possible. The significance of the remapping index was tested with a t-test against 0. The family-wise error rate was corrected for using a bootstrapping approach. Ten thousand iterations of t-tests on each subfield and city were performed on randomly permuted data. The distribution of t-values was then used to determine the corrected t-value at a given percentile.

## Region-of-interest MPS ANOVAs

Although we present only results from Left CA1 and Left CA3/DG in the body of the manuscript, CA1, CA3/DG, subiculum, and parahippocampal cortex (PHC) were demarcated from both hemispheres. Use of all hippocampal subfields in a region-of-interest (ROI) multivariate pattern similarity (MPS) analysis provided a complementary approach to the searchlight classifier. Searchlight techniques are limited in that they can demonstrate the location of information content, but generally do not support functional dissociations between regions. ROI approaches allow better access to investigate functional dissociations but suffer from the multiple comparison problem. To control for multiple comparisons here, we performed ANOVAs using all demarcated subfields to attempt to control for multiple comparisons.

First, we tested whether within city vs. between city neural similarity varied as a function of subfield, we performed a 2 (Within/Between) $\times$ 8 (Left and right, CA1, CA3/DG, Subiculum, and PHC) repeated-measures ANOVA. We found a main effect of subfield ($F_{(7,126)} = 8.4$, $p<0.001$) and an interaction effect of Within/Between city retrieval and subfield $F_{(7,126)} = 2.14$, $p = 0.04$. This suggested that the relationship of within/between city MPS varied by subfield. We then broke down within vs between for each city using the remapping index (*Figure 4*). Here an 8 $\times$ 4 subfield by city remapping repeated measures ANOVA revealed a main effect of subfield ($F_{(7,126)} = 2.835$, $p = 0.009$) suggesting that MTL subfields varied significantly in their tendency to "remap." These analyses suggested further investigation was warranted. This analysis is provided in the manuscript under "Classification of city-specific retrieval patterns in the hippocampus demonstrates successful decoding of all spatial contexts except the interfering environment" in the results section.

In the next section, we address incorrect vs correct retrieval of the interfering city. Here, we calculated MPS to compare correct and incorrect interference city trials with other city trials using matched visual stimuli (triads) during retrieval (see Experimental Procedures). An 8 $\times$ 4 subfield by condition repeated measures ANOVA revealed that the correlations between correct and incorrect trials of the different city comparisons (*Figure 5a,b* and *Figure 4—figure supplement 1* and *Figure 5—figure supplement 1*) varied as a function of subfield (significant interaction effect: $F_{(21,378)} = 2.5$, $p <0. 001$).

## Addressing possible confounds due to visual probes during retrieval

One possibility is that our analysis approach for Cities 1–3 or our presentation of novel stores for City 4 allowed for a visual confound. Thus, could our results be due to differences in visual features during retrieval rather than city-specific neural representations? Several lines of evidences argue

against this possibility. Cities 1–3 were perfectly visually matched in terms of what was presented to the participant and how we correlated these triads during our analyses. To provide more detail on the visual matching in our incorrect trial analysis using MPS (*Figure 5*) was perfectly visually matched (matched incorrect City 3 trials with corresponding City 1&2 trials vs. matched correct City 3 trials with corresponding City 1 & 2 trials), eliminating a visual confound as a possible counter interpretation. Thus, these findings for Cities 1–3 cannot be explained based on a trivial visual stimulus confound.

One unavoidable aspect is that our use of a truly novel city, City 4, did not involve visually matched triads as these stores were necessarily different than those in Cities 1-3. The pattern of findings based on a visual confound from City 4 alone, however, would have predicted a qualitatively and quantitatively different pattern of results. Specifically, a visual confound would predict higher classifier performance for City 4 and similar performance for Cities 1, 2, & 3. Our pattern of results, however, was that City 3 had lower classifier performance than Cities 1, 2, 4. Thus, the prediction provided solely by a visual confound was not supported by our data. Second, the visual confound would not predict the presence of a performance correlation with classification accuracy on City 3 (*Figure 3d*). Specifically, as we report in *Figure 3d*, classification performance improves with better individual subject retrieval performance on City 3. Thus, together, our paradigm and pattern of findings argue against a visual confound based explanation alone.

## Acknowledgements

The authors declare no conflict of interest. We thank Carol Barnes, Charan Ranganath, Lindsay Vass, and Aiden Arnold for helpful comments on the manuscript. The authors also wish to acknowledge support from NIH/NINDS grants NS076856 and NS093052 (ADE).

## Additional information

### Funding

| Funder | Grant reference number | Author |
|---|---|---|
| National Institute of Neurological Disorders and Stroke | NS076856 | Arne D Ekstrom |
| National Institute of Neurological Disorders and Stroke | NS093052 | Arne D Ekstrom |

The funders had no role in study design, data collection and interpretation, or the decision to submit the work for publication.

### Author contributions

CTK, Implemented the experiment, collected and analyzed the data, and wrote the manuscript.; JDS, Helped to write the manuscript.; JSL, Assisted with data collection.; ASH, Assisted with data analysis.; ADE, Conceived and designed the experiment and wrote the manuscript.

### Ethics

Human subjects: This study was approved by the Institutional Review Board at the University of California at Davis. Written informed consent was obtained from each participant before the experiment.

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
