## [Decision Letter]

Thank you for submitting your work entitled "Successful retrieval of competing spatial environments in humans involves hippocampal pattern separation mechanisms" for peer review at *eLife*. Your submission has been favorably evaluated by Timothy Behrens (Senior editor), a Reviewing editor, and two reviewers.

The reviewers have discussed the reviews with one another and the Reviewing editor has drafted this decision to help you prepare a revised submission.

The Reviewing editor and reviewers found the study interesting and valuable, and potentially a significant advance on what we know about neural representations of large-scale space in the human hippocampus and how they compete in memory. However, both reviewers had major concerns about the analyses and consequent interpretations of the data that can be addressed by additional analyses. Those concerns are listed below.

Reviewer #1:

My general assessment is that this is an interesting study and could be suitable for publication in *eLife*. The question is important and the methods used to address it are appropriate. The results are interesting and provide a potential advance. My major concern relates to the main findings, reported in Figure 3. The authors show evidence that Cities 1 and 2 are classified above chance. The authors interpret this is as being driven by differences in the pattern separation of internal representations of the Cities 1 and 2, since the stimuli are visually matched and motor responses and reversed across blocks. However, the difference between these two cites relates to a swap between the location of two stores, and I wondered whether the above chance decoding of these two cities is driven entirely by trials using these stores. This is important because two different mechanisms might be operating depending on the trial type. On trials where stores appear that did not contain swapped stores the stimuli and response is shared, there is no conflict in memory representation and above chance decoding on these trials I think would be in line with the authors main conclusion: the hippocampal regions separates out the two cities. This mechanism may also operate on the other types of trials containing the swapped stores, but in addition another mechanism may be operating which drives the classification – resolution of the conflict between the two responses before the response is made. The authors could attempt to address this by re-analyzing the data to test classification excluding trials in which the swapped stores were used, and also analyzing only those trials in which contained one or both swapped stores.

In the supplemental results, the authors should report classification and correlation measures from primary visual cortex if possible (depending on their scan orientation). This would help address whether sensory cortex also contains these patterns, which according to pattern separation theories, it should not.

Reviewer #2:

The manuscript presents an fMRI analysis of a city learning task that makes use of 4 cities with varying degrees of informational overlap. Cities 1 and 2 shared the greatest overlap and were identical except that the locations of a pair of landmarks were swapped. City 3 shared all landmarks with Cities 1 and 2, but they were arranged in a novel configuration. City 4 shared neither landmarks nor configurations with the other three cities. Participants learned the layout of each of the cities through navigation and map drawing in a behavioral session that occurred prior to the scan. During the scan, participants were tested using a 2 AFC judgment of the relative proximity of spatial landmarks to a reference landmark. Judgments were blocked by city. Behaviorally, the study found that performance for judgments in City 3 were impaired relative to all other cities (including Cities 1 and 2 which shared greater informational overlap). The authors used MVPA techniques to infer the properties of hippocampal neural representations that may help to explain the behavioral results. Using a searchlight and pattern-classification approach, they found that classifier accuracy was lower for City 3 than for other cities. They also found that incorrectly classified City 3 representations were classified as City 1 or City 2 representations at greater than chance (50%) levels and that classifier performance was correlated with behavioral performance. They also performed an ROI based pattern similarity analysis of the CA3/DG and CA1 hippocampal subfields where they found evidence of more distinct representations in CA3/DG than in CA1. They discuss the results in terms of pattern separation/completion and environment remapping. The manuscript presents a novel paradigm and analysis for investigating the retrieval of competing spatial environments. The searchlight/classifier analysis provides a link between voxel activity patterns and behavior. However, the ROI based MPS remapping analysis is less compelling and rests on somewhat shaky statistical evidence. The experiment and the results are interesting, but it's not clear that the evidence supports the claims made in the title and Abstract.

Major Concerns:

1) The arguments made and results reported by the manuscript converge upon the idea that the neural representations for Cities 1 and 2 are better separated from each other than City 3 is from Cities 1 and 2. Considering the fact that cites 1 and 2 share greater informational overlap with each other than they do with City 3, this result is somewhat surprising. Based upon the pattern-separation literature, one might expect to find greater representational overlap between Cities 1 and 2 due to the greater degree of overlapping information. The authors do not address this.

2) The classification analysis is the most compelling evidence presented in the manuscript. However, I am concerned that the deficits in classifier accuracy for City 3 could be due to a greater degree of error in the data-points presented to the classifier during training rather than being due to a property of the neural representation of City 3 itself. Based on the behavioral performance, it is clear that there are significantly fewer correct City 3 trials and therefore fewer "good" data-points available for classifier training (and concurrently more "bad" data-points) resulting in poor training of the classifier for City 3 relative to other cities. Further, the correlation between behavior and classifier accuracy could also be accounted for by an imbalanced training account. The classifier evidence could be made stronger by an analysis that accounts for discrepancies in the quality of training due to performance perhaps by restricting training to correct trials and capping the number of training trials for Cities 1, 2, and 4 to the number of correct trials in City 3.

3) The authors analyze the number of City 3 trials misclassified as City 1 or 2 trials, but do not present an analysis of the number of City 1 and City 2 trials misclassified as City 1-2 or City 3. Such an analysis may help build the case that City 1 and City 2 do indeed have non-overlapping neural representations and that the classifier isn't doing something like rejecting City 4 and classifying as City 1, 2 or 3 as a 3 sided die roll.

4) The Introduction sets up part of the goal of the experiment as differentiating between a pattern separation account of retrieval and an inhibition by executive control account, but never mentions the inhibition account again. The results are interpreted as evidence of pattern separation, but no argument is made as to why that interpretation is better than the inhibition account.

5) Results: "As predicted, we found that the slowest learning occurred for the interference city" – It's not clear from the preceding text why this would be the prediction.

6) Results, second paragraph: The authors suggest that the behavioral finding indicate that Cities 1 and 2 are more separable from each other than City 3, however Cities 1 and 2 are nearly identical meaning that the majority of the judgments would be the same between the two cities. It is possible that the participants could be retrieving a city 2 representation for a city 1 question and still coming to the correct judgment. The MVPA analysis provides better evidence for their claim, but the behavioral evidence alone does not suggest it.

7) The authors make use of pairwise post-hoc T-Tests to investigate main effects of ANOVAs, but do not report correcting for multiple comparisons.

8) In the subsection “ROI-based multivariate pattern similarity (MPS) voxel remapping suggests a functional dissociation of CA3/DG and CA1“, the authors parse a marginal interaction from and ANOVA by using single sample T-Tests against chance rather than by testing the relationships between the cities directly. There is no effect of city in the ANOVA and the marginal interaction is likely driven by city 4 in CA1. The paragraph seems to imply that remapping for City 3 within CA3/DG is lower than the other cities because the remapping score in that condition isn't different that 0 while the scores for 1, 2 and (arguably) 4 are different. This sort of analysis isn't statistically sound. It can be interesting to see if results differ from chance, but differences between conditions should be tested directly rather than through an intermediary. The authors should make an explicit account of the results of direct tests of the differences.

9) In the subsection “ROI-based multivariate pattern similarity (MPS) voxel remapping suggests a functional dissociation of CA3/DG and CA1“: use of a one-tailed t-test for city 4 remapping > 0 in CA1, but no justification provided.

10) In the subsection “Analysis of retrieval trials from the interference city suggests a partially unstable representation”, the authors analyze voxel pattern similarity in CA3/DG for City 1 and 2 using t-tests that check within city vs. between City 1&2 correlations, but do not correct for multiple comparison. Neither p-value would survive a correction, and the value for City 2 is marginal even without correction. This is weak statistical evidence to support the claim that Cities 1 and 2 have orthogonal representations.

---

## [Author Response]

Reviewer #1:

*My general assessment is that this is an interesting study and could be suitable for publication in* eLife*. The question is important and the methods used to address it are appropriate. The results are interesting and provide a potential advance. My major concern relates to the main findings, reported in Figure 3. The authors show evidence that Cities 1 and 2 are classified above chance. The authors interpret this is as being driven by differences in the pattern separation of internal representations of the Cities 1 and 2, since the stimuli are visually matched and motor responses and reversed across blocks. However, the difference between these two cites relates to a swap between the location of two stores, and I wondered whether the above chance decoding of these two cities is driven entirely by trials using these stores. This is important because two different mechanisms might be operating depending on the trial type. On trials where stores appear that did not contain swapped stores the stimuli and response is shared, there is no conflict in memory representation and above chance decoding on these trials I think would be in line with the authors main conclusion: the hippocampal regions separates out the two cities. This mechanism may also operate on the other types of trials containing the swapped stores, but in addition another mechanism may be operating which drives the classification – resolution of the conflict between the two responses before the response is made. The authors could attempt to address this by re-analyzing the data to test classification excluding trials in which the swapped stores were used, and also analyzing only those trials in which contained one or both swapped stores.*

We thank the reviewer for this suggestion. We’ve included results from both the swapped and stable trials using both a searchlight classification and ROI MPS analysis below. For the stable trial analysis, we limited City 1 and 2 trials to only those involving stores that remained fixed between Cities 1 and 2 (8 of the 20 stimuli). For the swapped trial analysis we limited City 1 and 2 trials of interest to only those involving at least one store that changed position between Cities 1 and 2 (12 of 20 stimuli). Both searchlight analyses result in a left hippocampal cluster with many of the same characteristics of our original searchlight analysis (see Figure 6 and Figure 7, see also Figure 3—figure supplement 1). Additionally, a region of interest multivariate pattern similarity analyses (MPSA) reveals converging results with no significant differences according to stable vs. swapped trials (Figure 8). Overall, these analyses suggest that it is not only the swapped but also the stable trials that elicit environment-specific hippocampal patterns during retrieval. We have added discussion of this issue to the Results as well as a supplementary figure addressing this issue.

Author response image 1.Stable City 1-2 trial classifier results. *p<0.05 Bonferrroni corrected, **p<0.01 Bonferroni corrected.**DOI:**
http://dx.doi.org/10.7554/eLife.10499.016

Author response image 2.City 1-2 swapped location trial classifier results. *p<0.05 Bonferrroni corrected, **p<0.01 Bonferroni corrected.**DOI:**
http://dx.doi.org/10.7554/eLife.10499.017

Author response image 3.Stable and swapped City 1 and 2 ROI analysis.**DOI:**
http://dx.doi.org/10.7554/eLife.10499.018

In the supplemental results, the authors should report classification and correlation measures from primary visual cortex if possible (depending on their scan orientation). This would help address whether sensory cortex also contains these patterns, which according to pattern separation theories, it should not.

We agree that a control region comparison is a sound idea; thank you for this suggestion. Unfortunately, due to the fact that our sequence specifically targets the medial temporal lobes using an oblique coronal acquisition sequence to increase signal to noise in this area, we were not able to look at classification within the visual cortex specifically (please see Ekstrom et al. 2009 for more details). However, we were able to expand our searchlight into areas surrounding the MTL including fusiform gyrus and inferior temporal cortex (ITC), where we would not expect significant remapping related effects. Consistent with predictions, we did not find clusters showing above chance environment classification when we expanded the searchlight into these areas. The largest cluster found in these areas was a 3 voxel cluster located in ITC, which was not large enough to pass cluster-size based family-wise error rate correction.

Reviewer #2:

Major concerns:

1) The arguments made and results reported by the manuscript converge upon the idea that the neural representations for Cities 1 and 2 are better separated from each other than City 3 is from Cities 1 and 2. Considering the fact that cites 1 and 2 share greater informational overlap with each other than they do with City 3, this result is somewhat surprising. Based upon the pattern-separation literature, one might expect to find greater representational overlap between Cities 1 and 2 due to the greater degree of overlapping information. The authors do not address this.

We agree that the degree of pattern separation demands might be expected to differ in some form for Cities 1 & 2 vs. City 3 (and vs. particularly City 4) based on their informational overlap and we did not find evidence, directly at least, for MPS being higher *between* Cities 1 vs. 2 vs. City 3/4. It is perhaps worth noting, however, that few studies have looked systematically at how pattern separation demands differ as a function of spatial input in humans, although one study in which we manipulated geometrical relatedness did suggest that neural pattern similarity was reduced as a function of more dissimilar geometric details (Stokes et al. 2015). For the cities in our current manuscript, however, we do not have a continuous measure of their relatedness. Thus, exactly how information differences will map onto neural differences is unclear. It is worth noting that several past studies in humans that have used fMRI to decode retrieval of recently learned episodic memories do in fact suggest distinct neural codes, even for partially overlapping memories (e.g., Chadwick et al. 2010). Consistent with these findings, it could be when there is no obvious continuous relationship between environments or stimuli, that pattern separation is more discrete in nature. This idea would be consistent with at least some proposals for CA3/DG function (i.e., Yassa et al. 2011). Additional evidence for these assertions is included in our first response to Reviewer 1 and Reviewer 2, points 3 and 6.

We have thus added the following text to the Discussion to attempt to address this issue:

“One potentially perplexing aspect of our findings is that we found similar degrees of remapping, and therefore putative neural pattern separation processes, for both the similar cities (Cities 1&2) and City 4, despite the fact that information content was significantly different for City 4 than Cities 1&2. […] Thus, one possible interpretation of our results is that pattern separation in CA3/DG may not always scale with behavioral details and thus may be a more discrete process that depends on the exact demands required by encoding and retrieval.”

2) The classification analysis is the most compelling evidence presented in the manuscript. However, I am concerned that the deficits in classifier accuracy for City 3 could be due to a greater degree of error in the data-points presented to the classifier during training rather than being due to a property of the neural representation of City 3 itself. Based on the behavioral performance, it is clear that there are significantly fewer correct City 3 trials and therefore fewer "good" data-points available for classifier training (and concurrently more "bad" data-points) resulting in poor training of the classifier for City 3 relative to other cities. Further, the correlation between behavior and classifier accuracy could also be accounted for by an imbalanced training account. The classifier evidence could be made stronger by an analysis that accounts for discrepancies in the quality of training due to performance perhaps by restricting training to correct trials and capping the number of training trials for Cities 1, 2, and 4 to the number of correct trials in City 3.

We thank the reviewer for this useful suggestion. We performed the recommended analysis by re-running the classifier but this time restricting the trials included during training such that each city was represented with an equal number of correct trials. As demonstrated in Figure 3—figure supplement 1, we see largely comparable results between analysis methods. While there is a reduction in the cluster size somewhat, which might be expected based on including fewer numbers of trials overall, we note the almost identical location within CA3/DG and CA1 and characteristics, including the significant correlation between City 3 classifier performance and retrieval performance. Thus, balancing trials such that each city contained equal numbers of trials overall, while reducing statistical power somewhat, did not change our overall results. We have added mention of this finding to the Results.

3) The authors analyze the number of City 3 trials misclassified as City 1 or 2 trials, but do not present an analysis of the number of City 1 and City 2 trials misclassified as City 1-2 or City 3. Such an analysis may help build the case that City 1 and City 2 do indeed have non-overlapping neural representations and that the classifier isn't doing something like rejecting City 4 and classifying as City 1,2 or 3 as a 3 sided die roll.

We thank the reviewer for the suggestion to better understand misclassification of City 1 & 2 trials. We plot the complete breakdown of all City 1 & 2 trials from Figure 3 in the manuscript (Figure 3—figure supplement 2). The pattern of results suggests that the classifier does not simply reject City 4 with high accuracy and classify Cities 1-3 at random. Instead, Cities 1 & 2 are incorrectly swapped by the classifier significantly less than chance (2^nd^ bar against chance t(18) = -3.9, p = 0.005 corrected). Additionally, Cities 1 & 2 are incorrectly classified as City 3 significantly less than chance (t(18) = -8.6,p<0.001 corrected). These findings are inconsistent with the idea that City 1-3 classification is random. Finally, and perhaps most importantly, the tendency for the classifier to misclassify Cities 1 & 2 as City 4 are not significantly different from chance (t(18) = 2.21, p = 0.16 corrected). Additionally, we note that Cities 1 & 2 are correctly classified more often than they are misclassified as City 4 (t(18) = 3.6, p = 0.013 corrected). We have added some discussion of these issues to the Results.

4) The Introduction sets up part of the goal of the experiment as differentiating between a pattern separation account of retrieval and an inhibition by executive control account, but never mentions the inhibition account again. The results are interpreted as evidence of pattern separation, but no argument is made as to why that interpretation is better than the inhibition account.

We agree that we discussed this insufficiently in the manuscript. We have now added the following to the Discussion:

“An alternative interpretation could be to attribute such errors to inhibition failure, for example, insufficient inhibition of City 1&2 representations by prefrontal cortex could lead to those being erroneously retrieved when attempting to retrieve City 3. […]. Thus, our findings overall support the importance of pattern completion and separation, particularly in CA3/DG, to spatial remapping and appear less easily reconciled with an inhibition-based account.”

5) Results: "As predicted, we found that the slowest learning occurred for the interference city" – It's not clear from the preceding text why this would be the prediction.

We have now tried to clarify in the Results why we predicted the slowest learning for City 3:

“We predicted that City 3 would experience slower learning relative to Cities 1 & 2 because all of City 3’s store locations were in conflict with store locations from Cities 1 & 2. Cities 1 & 2, on the other hand, had only 2 conflicting store locations, which could be learned via a simple swap.”

6) Results, second paragraph: The authors suggest that the behavioral finding indicate that Cities 1 and 2 are more separable from each other than City 3, however Cities 1 and 2 are nearly identical meaning that the majority of the judgments would be the same between the two cities. It is possible that the participants could be retrieving a city 2 representation for a city 1 question and still coming to the correct judgment. The MVPA analysis provides better evidence for their claim, but the behavioral evidence alone does not suggest it.

We thank the reviewer for suggesting clarification on this issue. Although we did not present thorough evidence for behavioral separation between Cities 1 and 2 in the previous version of the manuscript, we do believe that our data supports this assertion. We designed our stimuli such that trials involving stores that swapped locations between Cities 1 and 2 were over-represented. For instance, 12 of 20 trials involved at least one swapped store between Cities 1 & 2 and 9 of these 12 had a different correct response in City 1 vs. City 2. Therefore, a subject could score a maximum of 55% accuracy in City 2 based on knowledge of City 1 alone, and vice versa and all subjects were well above this threshold. Additionally, even if we assume that subjects correctly answer all trials involving the same response in Cities 1 & 2 and guess on trials involving different correct responses, we would expect chance level accuracy of 77.5% for Cities 1 & 2. All but 2 subjects had accuracy above 77.5% for *both* Cities 1 and 2 (one scored 77.5% on City 1 and 92.5 percent on City 2; the other scored 72.5 and 57.5 on Cities 1 & 2, respectively). Furthermore, taking the lower performance for Cities 1 and 2 for each subject and testing the result against 77.5%, subjects still performed significantly above chance (t(18) = 5.4, p<0.0001). Thus, based on both the retrieval data (collected during imaging) and the map drawing data presented in the supplemental figures, we believe that we have provided relatively strong behavioral evidence for Cities 1 and 2 do not involve the same underlying representation.

7) The authors make use of pairwise post-hoc T-Tests to investigate main effects of ANOVAs, but do not report correcting for multiple comparisons.

We have corrected this error and now use family-wise error correction where appropriate.

8) In the subsection “ROI-based multivariate pattern similarity (MPS) voxel remapping suggests a functional dissociation of CA3/DG and CA1“, the authors parse a marginal interaction from and ANOVA by using single sample T-Tests against chance rather than by testing the relationships between the cities directly. There is no effect of city in the ANOVA and the marginal interaction is likely driven by city 4 in CA1. The paragraph seems to imply that remapping for City 3 within CA3/DG is lower than the other cities because the remapping score in that condition isn't different that 0 while the scores for 1, 2 and (arguably) 4 are different. This sort of analysis isn't statistically sound. It can be interesting to see if results differ from chance, but differences between conditions should be tested directly rather than through an intermediary. The authors should make an explicit account of the results of direct tests of the differences.

We apologize regarding the confusion on this point. The t-tests against zero were not intended as a post-hoc test of the ANOVA. Rather, these tests were included because significance above zero would indicate significant remapping. We have revised our presentation of these results and have taken additional measures to correct for family-wise error. We have also deleted any post-hoc t-tests that were not specifically corrected for multiple comparisons.

9) In the subsection “ROI-based multivariate pattern similarity (MPS) voxel remapping suggests a functional dissociation of CA3/DG and CA1“: use of a one-tailed t-test for city 4 remapping > 0 in CA1, but no justification provided.

We have corrected this error and clarified our approach in the Methods and Results sections.

10) In the subsection “Analysis of retrieval trials from the interference city suggests a partially unstable representation”, the authors analyze voxel pattern similarity in CA3/DG for City 1 and 2 using t-tests that check within city vs. between City 1&2 correlations, but do not correct for multiple comparison. Neither p-value would survive a correction, and the value for City 2 is marginal even without correction. This is weak statistical evidence to support the claim that Cities 1 and 2 have orthogonal representations.

We agree with this concern. We now perform a resampling-based multiple comparisons correction for these comparisons based on the family-wise error rate to test whether remapping scores are significantly above zero. Bootstrapping was advantageous because it does not make any assumptions about the underlying variance or covariance structure of the data. We performed 1000 iterations of randomly resampling the pool of all within and between-city MPS scores for CA3 and CA1 and t-testing these values against 0. This generated a null t-distribution representing the family-wise error rate. We then computed the 95% percentile of the null t-distribution to yield a p<0.05 corrected t threshold of 1.76 (one tailed t-test). Thus, t-values exceeding this value would suggest that remapping values were greater than zero, or that within environmental neural patterns were greater than between environmental patterns. We have corrected this issue throughout the Results section that deals with our multivariate findings.